# Exploring the Aβ Plaque Microenvironment in Alzheimer’s Disease Model Mice by Multimodal Lipid-Protein-Histology Imaging on a Benchtop Mass Spectrometer

**DOI:** 10.3390/ph18020252

**Published:** 2025-02-13

**Authors:** Elisabeth Müller, Thomas Enzlein, Dagmar Niemeyer, Livia von Ammon, Katherine Stumpo, Knut Biber, Corinna Klein, Carsten Hopf

**Affiliations:** 1Center for Mass Spectrometry and Optical Spectroscopy (CeMOS), Mannheim University of Applied Sciences, 68163 Mannheim, Germany; e.mueller@hs-mannheim.de (E.M.); t.enzlein@hs-mannheim.de (T.E.); l.vonammon@hs-mannheim.de (L.v.A.); 2Medical Faculty Heidelberg, Heidelberg University, 69120 Heidelberg, Germany; 3Bruker Daltonics GmbH & Co. KG, 28359 Bremen, Germany; 4Bruker Scientific, LLC, Billerica, MA 01821, USA; 5AbbVie Deutschland GmbH & Co. KG, 67061 Ludwigshafen, Germany; 6Mannheim Center for Translational Neurosciences (MCTN), Medical Faculty Mannheim, Heidelberg University, 68167 Mannheim, Germany

**Keywords:** MALDI mass spectrometry imaging, lipid, HiPLEX immunohistochemistry, Alzheimer’s disease, microglia, amyloid plaque

## Abstract

Amyloid-β (Aβ) plaque deposits in the brain are a hallmark of Alzheimer’s disease (AD) neuropathology. Plaques consist of complex mixtures of peptides like Aβ_1–42_ and characteristic lipids such as gangliosides, and they are targeted by reactive microglia and astrocytes. **Background:** In pharmaceutical research and development, it is a formidable challenge to contextualize the different biomolecular classes and cell types of the Aβ plaque microenvironment in a coherent experimental workflow on a single tissue section and on a benchtop imaging reader. **Methods:** Here, we developed a workflow that combines lipid MALDI mass spectrometry imaging using a vacuum-stable matrix with histopathology stains and with the MALDI HiPLEX immunohistochemistry of plaques and multiple protein markers on a benchtop imaging mass spectrometer. The three data layers consisting of lipids, protein markers, and histology could be co-registered and evaluated together. **Results:** Multimodal data analysis suggested the extensive co-localization of Aβ plaques with the peptide precursor protein, with a defined subset of lipids and with reactive glia cells on a single brain section in APPPS1 mice. Plaque-associated lipids like ganglioside GM2 and phosphatidylinositol PI38:4 isoforms were readily identified using the tandem MS capabilities of the mass spectrometer. **Conclusions:** Altogether, our data suggests that complex pathology involving multiple lipids, proteins and cell types can be interrogated by this spatial multiomics workflow on a user-friendly benchtop mass spectrometer.

## 1. Introduction

Alzheimer’s disease (AD), a progressive neurodegenerative disorder, is the leading cause of dementia in the elderly population that affects over 50 million people globally. This number is expected to rise as populations age [1,2]. AD is characterized by cognitive decline, memory loss, and behavioral changes. Key pathological features include extracellular amyloid-beta (Aβ) plaques, intracellular neurofibrillary tangles (NFTs) of hyper phosphorylated tau, the loss of neurons, and, more recently, neuroinflammation driven by reactive microglia and astrocytes as a potential further AD hallmark [3]. Aβ plaques in AD primarily consist of Aβ peptides, formed by the abnormal cleavage of amyloid precursor protein (APP) by β- and γ-secretase enzymes [4]. The cellular and molecular microenvironment around Aβ plaques is complex, and it influences disease progression. Immune cells like microglia and astrocytes cluster around plaques, initially clearing Aβ, but eventually become dysfunctional with chronic responses, e.g., releasing pro-inflammatory cytokines [5]. Altered lipid metabolism, especially of gangliosides, in the vicinity of plaques may contribute to further disease progression [6]. Several mouse models have been genetically engineered to replicate key AD features, such as Aβ plaques, tau pathology, and, in some cases, cognitive decline. Transgenic models, like APPPS1 mice, overexpressing human genes linked to early-onset familial AD (e.g., mutant APP and presenilin 1 (PS1)) are used in pharmaceutical R&D to study Aβ pathology and plaque formation [7]. Current AD treatments, like the FDA-approved immunotherapy drugs for patients with early Alzheimer’s disease (Lecanemab, Donanemab), can significantly slow down cognitive decline [8,9]. Although encouraging, a lot of research is still needed for the effective treatment of the disease. However, advances in genomics, biomarkers, and imaging have offered new insights into potential therapeutic targets, thus fueling optimism for new drug classes [10,11]. New technologies for pharmaceutical inquiry have become available as well.

Matrix-Assisted Laser Desorption/Ionization Mass Spectrometry Imaging (MALDI MSI) is an advanced technique used to visualize the spatial distribution of different bimolecular classes in tissue sections coated with a chemical matrix. MALDI MSI enables the creation of molecular maps of tissue, allowing visualization without the need for labels or dyes [12]. It detects a wide range of biomolecules, from small metabolites to proteins, depending on the matrix and conditions [12]. It can offer high spatial resolution, typically from tens to hundreds of micrometers [13]. Technological advances are now enabling studies even at cellular levels [14]. MALDI MSI has been widely used in AD research to investigate the brain’s molecular landscape [6,15,16]. Multiple studies have focused on mapping the spatial distribution of Aβ peptides from brain tissue sections, thereby enabling studies of plaque location, composition, and differentiation between various Aβ isoforms [17]. MALDI MSI has been used to visualize Aβ plaques in transgenic mouse models and human AD patient samples, revealing that the distribution of Aβ species correlates with other pathological features such as astrocytes, as well as lipid species [6,16,18].

Biochemical and clinical studies indicate that, alongside Aβ and tau pathology, alterations in neuronal lipid metabolism are involved in AD progression [19]. The ε4 allele of the apolipoprotein E-encoding gene (APOE) is one of the most important genetic risk factor in AD [20]. In recent years, genome-wide association studies (GWAS) have identified more lipid-associated genes as risk factors, like the lipid transporter TREM2, highlighting the potential of lipid metabolism as new therapeutic target [21]. Current research focuses on identifying region-specific lipid changes in AD brains compared to healthy controls, particularly in areas like the hippocampus and cortex which are affected by plaque depositions in the progression of the disease [6,15]. The accumulation of distinct lipid species, such as glycerophospholipids and sphingolipids like gangliosides were identified in the vicinity of Aβ plaques in a transgenic mouse model and human AD patient samples using MSI [6,15,16]. Some lipids, such as gangliosides, interact with Aβ peptides and promote their aggregation into plaques [22]. The metabolism of other lipid classes, such as derivatives of phosphatidylinositol, seem to be disrupted by Aβ oligomers [23]. The increase in ceramides can be related to the observed decrease in sphingomyelin (namely sulfatides). This aligns with discussions in the context of AD, where the degradation of sulfatides is thought to occur during the disease progression [24].

To comprehensively assess the complexity of AD, it is fundamentally important to consider as many classes of biomolecules as possible [25]. For years, large molecules such as amyloid-beta peptides and phosphorylated tau have gained most of the attention in AD research [26]. Recently, there is a growing interest in the metabolic activities of cells in the plaque microenvironment [6,16]. Ideally, the analysis of lipids, peptides, and proteins would be accomplished on a benchtop imaging reader from one single tissue sample. However, while the analysis of lipids is very common using MALDI MSI, the detection of large molecules, such as intact proteins, is more challenging [27]. Strategies like the on-tissue digestion of proteins result in very complex data sets with high background, which are limited to the detection of highly abundant peptides and provide very low sequence coverage of proteins [28,29]. Typically, proteins of interest are investigated by immunohistochemistry (IHC), using one or even multiple (high-plex) antibody labels on the tissue sample [30,31]. Consequently, combining lipid imaging and protein imaging from one single tissue slide presents a technical challenge.

Here, we demonstrate the feasibility of the multiomic and multimodal MALDI MSI of lipids, protein markers, and H&E histology, all on the same tissue section and on a benchtop MALDI imaging instrument. Using MALDI HiPLEX-IHC technology [32] to visualize amyloid-β plaques and known cell/protein markers of their pro-inflammatory cellular microenvironment, we contextualize lipids as key components of the plaque microenvironment in an APPPS1 mouse brain sample. Finally, we demonstrate the ability to elucidate lipid structures directly on tissue with MS2 fragmentation analysis, all on the same instrument.

## 2. Results

### 2.1. Workflow for Multimodal Lipid MSI, MALDI HiPLEX Immunohistochemistry (IHC), and H&E Histology on a Benchtop Mass Spectrometer

In this study, we aimed to develop a new workflow for multimodal lipid MSI and MALDI HiPLEX immunohistochemistry (IHC) on the same tissue section. Our goal was to achieve this using a newly available benchtop MALDI-TOF/TOF mass spectrometer. The experimental order for MSI analysis and staining was designed to optimize signals for all components, although the data analysis process did not follow that order. To this end, a single cryosection of fresh-frozen APPPS1 mouse brain or a wild-type control were spray-coated with the vacuum-stable MALDI matrix DMNB-2,5-DHAP [33] for untargeted lipid MS imaging for *m*/*z* 400−1600 in negative ionization mode. Subsequently, the MALDI matrix was removed for further processing. For MALDI HiPLEX-IHC on the same sample section, the tissue was fixed in paraformaldehyde and washed to remove lipids and metabolites. The brain section was then incubated with multiple antibodies simultaneously. Each antibody was conjugated to a distinct photo-cleavable peptide mass tag (PCMT) as a unique reporter for a single protein marker. After the staining and UV-cleavage of mass tags from the antibodies, the sections were spray-coated with the MALDI matrix CHCA, followed by a second MS imaging run in the positive ionization mode to record the PCMT reporters. This allowed us to acquire targeted peptide mass tag data for *m*/*z* 600−4000 (Figure 1a) in addition to the lipid MSI data.

For the correlative investigation of multimodal data, the two MSI data sets and the H&E image obtained from the same slide were co-registered using in-house M^2^aia software [34,35]. This allowed for the visualization of all the three layers (H&E, lipids, and protein HiPLEX-IHC) to be possible (Figure 1b). For data analysis, we reversed the order used for data acquisition (lipids first, then HiPLEX-IHC) and focused on the spatial analysis of protein markers first.

### 2.2. MALDI HiPLEX-IHC Staining Revealed AD Biomarker Co-Localization in Microenvironment of Aβ_1–42_-Positive Plaques in APPPS1 Mouse Brain

The focus of Alzheimer’s disease (AD) has expanded beyond Aβ plaques and hyper-phosphorylated tau proteins. It now includes additional molecular characteristics closely associated with Aβ deposits. The testable hypothesis is that Aβ peptide deposition, tau pathology, and reactive immune cells such as microglia collectively drive neurodegeneration in AD and ultimately lead to dementia [36]. This underscores the importance of uncovering the microenvironment of Aβ plaques to fully understand the progression of the disease. Using the new workflow on a benchtop mass spectrometer, we investigated the surroundings of Aβ_1–42_-positive plaques using multiple PCMT-labeled antibodies to localize multiple neurology markers in one APPPS1 mouse brain section. An H&E image provided tissue morphology information, including the cortex and hippocampal regions as landmarks within the mouse brain tissue sample (Figure 2a, Appendix A) [37]. Structural markers such as NeuN included in the HiPLEX antibody panel supported navigation of relevant brain areas, e.g., the pyramidal cell layer of dentate gyrus in the hippocampus (Figure 2b, Appendix A). We detected many dense Aβ plaques using an antibody directed against Aβ_1–42_ (ion image of its reporter mass tag at *m*/*z* 1771.5; Figure 2c) in various brain areas including the cerebral cortex and hippocampus, in accordance with other studies of the APPPS1 mouse using other imaging technologies [7]. We labeled fifty of this Aβ_1–42_ plaques in red and displayed this plaque ROIs for all further ion images. No Aβ_1–42_-positive regions were observed in the WT mouse sample (Appendix A). Microglia were detected with the anti-Iba-1 antibody (Figure 2d; single ion image of *m*/*z* 960.1 reporter tag) with high expression in the hippocampus and cortex of 13-month-old APPPS1 mice but not in 14-month-old WT mice (Appendix A). Furthermore, we investigated the astrogliosis marker of glial fibrillary acidic protein (GFAP, *m*/*z* 1011.9), which displayed a distribution pattern similar to that of Iba-1. Additionally, there was some visualization in the transitions of the hippocampal sub-regions (Figure 2e). For unknown reasons, this distribution was also detected in the WT sample (Appendix A). As a third marker, we investigated the localization of amyloid precursor protein (APP, *m*/*z* 1723.6) from which Aβ peptides are derived from. The spatial distribution of this marker also showed partial signal enrichment in the cortex and hippocampus. Furthermore, we observed a presumably endogenous signal pattern of unknown origin in the outer hippocampus area (Figure 2f). This signal pattern was also observed in the WT sample to some extent (Appendix A).

To further investigate the correlation between Aβ_1–42_-positive plaques and the three individual neurology markers, we superimposed each of these ion images on that of Aβ_1–42_ (Figure 2g–i). The overlay of the Aβ_1–42_ signal (cyan) on that of Iba-1 (pink; microglia) revealed a co-localization of 84.8 ± 13% both in the cortex and hippocampal regions (white, Figure 2g, Appendix A). The white filled arrow and arrowhead indicate an example Aβ_1–42_ cluster in the hippocampus with co-localizing Iba-1 (Figure 2c,d,g). We further explored ongoing astrogliosis in close vicinity to plaque material by overlaying the ion images of Aβ_1–42_ with the GFAP mass reporter. As for the microglia, we detected multiple plaque sites co-localizing with GFAP (29.6 ± 9% co-localization, indicated by white filled arrow and filled arrowhead, Appendix A). However, we detected high levels of GFAP-positive signals with large clusters in the hippocampus, while there seemed to be fewer signals co-localizing with Aβ_1–42_ in the cortex (Figure 2h). Finally, in the APPPS1 mouse brain section, the overlay of Aβ_1–42_ (cyan) and APP (orange) revealed a co-localization of 69.8 ± 18% in the cortex and hippocampus (Figure 2i, Appendix A). For most Aβ_1–42_-positive stained regions, APP appeared to co-localize (indicated by white filled arrow). Nevertheless, some Aβ_1–42_-positive plaques did not have APP in their surroundings (indicated by white filled arrowhead).

The intact protein imaging and co-localization of different neurological markers with Aβ_1–42_-positive plaque regions suggested a difference in the molecular composition of the regions of interest (ROIs). We therefore further investigated whether the composition around the Aβ plaques also differed at the lipid level.

### 2.3. Multimodal MSI Revealed Lipid Composition of Individual Aβ_1–42_-Positive Microglia Regions in APPPS1 Mouse Brain Tissue

Altered lipid homeostasis in the brain of AD patients has become increasingly interesting, as multiple GWAS have identified potential AD risk genes associated with lipid metabolism (reviewed in [21]).

We therefore extended the analysis of key protein markers in the Aβ plaque environment and combined it with the previously recorded lipid data layer using untargeted MALDI lipid imaging. We investigated the co-localization of specific lipid species with Aβ_1–42_-positive plaque regions in the APPPS1 mouse brain to identify potential lipid markers.

The MALDI MSI of lipids was performed first on the tissue section and was followed by MALDI HiPLEX-IHC. Untargeted lipid data were acquired in a mass range of *m*/*z* 400–1600 using a vacuum-stable lipid matrix [33]. MALDI HiPLEX-IHC, MALDI MS lipid data, and the respective H&E image of the tissue section were co-registered. ROIs were defined by the Aβ_1–42_-reporter *m*/*z* 1771.5 (Figure 3a(i)). In total, fifty regions of Aβ_1–42_-positive areas in the cortex and hippocampus were manually selected from the MALDI HiPLEX-IHC data set (Figure 3a(ii)). Aβ_1–42_-containing ROIs could be displayed as an overlay on the H&E image for regional overview (Figure 3a(iii) and on single ion images of the lipid features (Figure 3b(i–viii)).

The visualization of single-ion intensity distributions for individual lipid species showed a consistent deposition-like pattern throughout the cortical and hippocampal regions. We were able to show accurate correlations of lipid features such as glycosphingolipids with the Aβ_1–42_-positive plaque regions. The inspection of these ion images revealed the plaque-associated accumulation of gangliosides (GM3 36:1, *m*/*z* 1179.8; GM3 38:1, *m*/*z* 1207.8; GM2 36:1, *m*/*z* 1382. 9; GM2 38:1, *m*/*z* 1410.9; Figure 3b(i–iv)) in the APPPS1 brain. No substantial elevation in gangliosides was found in the control sample (Appendix A). The difference in GM3 36:1 between the APPPS1 and WT brains was significant in both the cortex and hippocampal regions (Appendix A). GM3 d38:1, GM2 d36:1, and GM2 d38:1 showed significant enhancement in the cortex but not in the hippocampal region of the APPPS1 mouse brain (Appendix A). This accumulation, particularly in the cortex, aligns with previous studies that have shown the accumulation of gangliosides in amyloid plaques in a transgenic mouse model [15] as well as in human brain samples [6,16]. In this study, we did not detect any significant differences in GM1 between the APPPS1 and WT samples (Appendix A). In addition to gangliosides, we detected certain phosphatidylinositol species with comparable accumulation pattern in the APPPS1 mouse brain. The feature *m*/*z* 559.4 putatively belonging to lysophosphatidylinositol (LPI) 18:0 showed significantly higher intensity in the cortex and hippocampal regions of the APPPS1 mouse samples in comparison to the WT mouse section (Figure 3b(v), Appendix A). Additionally, *m*/*z* 855.6 belonging to phosphatidylinositol 38:4 showed increased intensity in the ROIs defined by Aβ_1–42_ based on the ion image (Figure 3b(v,vi)). However, there was no significant difference between the APPPS1 and WT mouse brain samples, neither in the cortex nor in the hippocampus (Appendix A). Whereas LPI 18:0 seemed to primarily appear in the vicinity of putative plaques, PI 38:4 was additionally located in the pyramidal layer of the hippocampus and, to a certain extent, in the cortex. Another lipid class that has been shown to have altered behavior in the APPPS1 mouse model is sulfoglycosphingolipids (STs), especially sulfatides [16]. However, compared to other lipid classes, gangliosides did not appear to accumulate in plaque regions. Rather, they appeared to be less intense in the immediate vicinity of the Aβ_1–42_-positive plaques [16]. When we assessed the depletion of sulfatides in Aβ_1–42_-positive plaques, we indeed saw a drop in intensity in those areas compared to the control areas in the WT mouse but not to a significant degree (Figure 3b(vii,viii), Appendix A). Additionally, their spatial distribution indicated that they were most intense in the white matter of both the APPPS1 and WT brains (Figure 3b(vii,viii), Appendix A).

To investigate potential differences in the lipid composition of individual plaques, we focused on three ROIs. The white filled arrowhead pointing to an Aβ_1–42_-positive region in the cortex defines the first region (Figure 3a(i)). The same ROI in the lipid data set shows the accumulation of gangliosides (i–iv), (lyso)-phosphatidylinositol (v,vi) as well as a potential depletion of the two sulfatides (vii,viii). The white filled arrow indicates an Aβ_1–42_-positive plaque ROI as well as only ganglioside and lyso-phosphatidylinositol (LPI 18:0) accumulation in the hippocampal region (Figure 3b). The white unfilled arrows indicate lipid deposition in the hippocampus which was apparent for GM3 d36:1 and GM2 d36:1 but not for GM3 d38:1 and GM2 d38:1.

### 2.4. Structural Elucidation of Candidate Lipid Biomarkers by TOF/TOF On-Tissue Fragmentation Using Benchtop Mass Spectrometer

Our analysis demonstrates that, not only for protein markers but also on the lipid level, differences in the microenvironment of individual Aβ plaque regions appeared in the APPPS1 mouse sample. To report accurate information on these potential lipid markers, a lipid TOF/TOF fragmentation analysis study was performed on the same benchtop MS to accurately identify the lipid species.

After the acquisition of the MALDI MSI lipid data, we performed the manual selection of precursors for on-tissue fragmentation to identify and annotate the candidate lipid biomarkers described in the previous section.

The ganglioside GM2 (36:1) (C_67_H_121_N_3_O_26_; *m*/*z* 1383.8 [M]) is an example of a candidate marker that we successfully fragmented (Figure 4a). Chemical reference structures were obtained from LIPIDMAPS [38]. The intense precursor at *m*/*z* 1383.0 [M-H]^−^ was of a high intensity, meaning that the fragmentation efficiency could be further improved if the MS^2^ was ambiguous. The peak at *m*/*z* 1091.8 corresponded to the [GalGalGlcCer-H]^−^ ion without the sialic acid linkage and the fragment at *m*/*z* 290.0 displayed the counterpart, specifically the N-acetylneuraminic acid linked to the inner Gal. The ion assignment was in accordance with published recommendations [39]. However, we could not detect any fatty acid fragments or the sphingolipid itself. Therefore, we cannot report on the exact arrangement and length of the respective fatty acyl chains (Figure 4a, bottom).

Similarly, the structure of phosphatidylinositol 38:4 (18:0/20:4) (C_47_H_83_O_13_P; *m*/*z* 886.5 [M]) was determined. The fragmentation spectrum resulted in a peak at *m*/*z* 581.3 which corresponded to the triglyceride backbone connected to the inositol phosphate head group. The fragments *m*/*z* 303.1 and *m*/*z* 283.2 indicated the lengths of two fatty acid chains; however, we have no additional information to report on exact double bond positions. The fragments *m*/*z* 241.0, *m*/*z* 153.0, and *m*/*z* 96.9 were indicative of the head group components (Figure 4b).

The on-tissue fragmentation of ten lipid *m*/*z* features was performed with success (Appendix A).

## 3. Discussion

Multimodal analysis for pathological conditions such as AD are urgently needed to gain comprehensive insight into the various molecular and cellular processes of the disease (discussed in [40]). MSI offers the capability to analyze a wide range of small molecules and reveal their spatial distribution within tissue [12]. Combining this technique with HiPLEX-IHC enables the analysis of many relevant biomolecules from fatty acids to multi-kDa proteins. This study demonstrates that all measurements required for multimodal analysis can be performed on a single benchtop mass spectrometer.

We successfully obtained data for lipids, crucial protein biomarkers, and histological information from a single tissue section derived from APPPS1 mice and a WT control. This study showed the co-localization of different protein biomarkers in the microenvironment of Aβ_1–42_-positive plaques in an APPPS1 brain sample. We detected a high load of Aβ_1–42_-positive plaques in the cortex and hippocampus and showed the accumulation of two important immune markers (Iba-1 and GFAP) at the plaque sites using photo-cleavable mass tag reporters. Additionally, we detected increased APP expression in the vicinity of plaques in the APPPS1 mouse in comparison to the WT sample. Our findings suggest that co-localization between Aβ_1–42_ plaques and all three neural protein markers was not absolute, revealing noticeable differences in the molecular composition of the microenvironment of individual plaques. Upon further investigation of the lipid composition of the Aβ_1–42_-positive plaques, distinct lipid classes were accumulated in the ROI.

This study investigated Aβ plaques and activated microglia in the same tissue section. Microglia are the resident immune cells of the central nervous system and play a key role in brain homeostasis and neuroinflammation (reviewed in [41]). They have been observed in close association with amyloid-beta aggregates and are now considered a potential third hallmark of AD alongside Aβ plaques and tau tangles [42]. In the recorded ion images, overlays, and statistical analysis of the Aβ_1–42_ and Iba-1 signals, we observed a high degree of co-localization of plaques and potentially clustered microglia in the cortex and hippocampus. We suggest that this indicated the recruitment of microglia cells around amyloid-beta. This observation is consistent with the recent literature showing the accumulation of Iba-1 positive microglia in APPPS1 mice and human AD patients [43,44]. The question of whether plaque-associated microglia have a protective effect or contribute to the progression of the disease cannot be answered here. Earlier studies have shown that microglia play a crucial role in AD, as they respond to Aβ material by attempting to clear accumulations by phagocytosis or degrading enzymes [45]. Clustered microglia surrounding Aβ plaques are mainly responsible for the phagocytosis of fibrillary Aβ peptides in APPPS1 mice [46]. Thereby, microglia change from a resting into a reactive state that releases pro-inflammatory molecules to further drive neuro-inflammation [46].

There is further support for the premise of enhanced neuro-inflammation from astrocytes in AD patient brains. They have been observed as plaque components in AD animal models and human AD patient samples [5,47]. GFAP is the main marker for astrogliosis, a process during which astrocytes transition into a reactive state during inflammation [48]. In this study, we identified GFAP-positive regions near Aβ_1–42_-positive plaques. This result aligns with previous research on reactive astrocytes surrounding plaques in APPPS1 mice [49]. However, our results suggest that not all detected Aβ_1–42_ plaques were accompanied by GFAP-positive astrocytes. This raises the question of whether the astrogliosis response might have varied across different areas of the affected brain areas. In various animal tissue samples and human patient samples, multiple laboratories have reported differing results, including a complete lack of astrogliosis. The absence of GFAP-positive astrocytes could indicate an impaired astrogliotic response, possibly due to immune system failure in the later stages of plaque development (further discussed in the review of [47].

Amyloid precursor protein (APP), a ubiquitously expressed transmembrane protein, plays a central role in AD as a precursor of amyloid peptides like Aβ_1–42_. These Aβ peptides aggregate and form Aβ plaques [4]. In this study, we observed accumulating APP signals mainly in the cortex and hippocampus, mostly co-localizing with Aβ_1–42_-positive plaques. The APPPS1 mouse model is genetically engineered to overexpress human APP including familiar AD mutations, resulting in increased levels of APP in the brains of these mice [7]. Therefore, the apparently clustered APP at the plaque site in the transgenic mouse could be an artifact of this AD model. This suggests that APP levels might not be increased or clustered to the same extent at the plaque site in models like the APP^NL-G-F^ mouse model which expresses APP at wild-type levels or in samples from patients with sporadic late-onset AD [50]. Nevertheless, the Selkoe group tested different antibodies directed against APP in human AD patient samples and suggested the presence of full-length APP in some but not every senile Aβ plaque [51]. In our MALDI-IHC experiments, APP staining revealed positive APP signal intensity in the pyramidal layers of CA and the granule layer of the dentate gyrus (DG). The Zheng group examined the localization of endogenous APP in a mouse brain and found that APP was specifically expressed in NeuN-positive neurons [52]. This result could explain our finding of increased APP-positive signal intensity in the APPPS1 and WT samples in this exact anatomical region, which seemed very similar to the NeuN signal distribution in the hippocampal area of the APPPS1 and WT mouse sections.

In addition to protein markers, lipids have recently received increased attention in AD research. The transcriptomic signature of lipopolysaccharide (LPS)-activated microglia has revealed changes in lipid- and lipoprotein-associated genes like ApoE and TREM2 [53], suggesting a strong link between microglial activation, lipid metabolism in the brain, and potentially the neurodegenerative progression of the disease. Therefore, we investigated different lipid species using untargeted MSI.

In agreement with other animal and human studies, we detected GM, PI, and ST species in close association with Aβ pathology [6,15,16]. Gangliosides represent a subclass of cell membrane glycosphingolipids. They are major components of lipid rafts in neuronal membranes and therefore essential for cell signaling processes and maintaining the activity of protein–protein and protein–lipid interaction [54]. In AD, there is increasing evidence of altered ganglioside metabolism [22]. We detected increased levels of GM2 and GM3 species in the APPPS1 mouse model in the cortex and hippocampus, thus confirming earlier MSI studies [6,15].

In our lipid measurements of the APPPS1 mouse brain, we did not detect significant accumulations of GM1 in the surrounding of Aβ_1–42_-positive plaques. We note that the endogenous intensity of GM1 in the WT mouse was widely distributed. Compared to that, the intensity of GM1 in the APPPS1 model did not show any significant differences (Appendix A). A recent study came to the same conclusion [55]. The high endogenous GM1 intensity could be explained by complex gangliosides (such as GM1) being the predominant form in the brain [55]. However, recent published data on a different transgenic mouse model (tgArcSwe) and human familial and sporadic AD patient samples showed GM1 as an enriched species in the Aβ plaque-like regions of interest [6,16]. Both mouse models and human data clearly demonstrate an increase in less complex ganglioside species like GM2 and GM3 [6,15,16]. It is possible that low levels of GM1 in the APPPS1 mouse brain and increased levels of GM2 and GM3 could indicate an altered lysosomal degradation of gangliosides [56]. GM1 might be more extensively degraded than other gangliosides, resulting in more accumulation of GM2 and GM3 [56]. The accumulation of certain GMs in the brain has also been identified as a pathological feature of lysosomal lipid storage disorders [57]. This lipid accumulation results from the loss of the function of key enzymes involved in GM metabolism. Even so, individuals affected by this disorder may die at very young ages; researchers have detected amyloid-β peptide accumulations similar to the ones found in AD patients [57,58]. This might indicate a connection between lipid storage disorders and AD, and research groups like the Nordstrom lab have already started testing on a mouse model of AD [59]. Overall, the precise role of gangliosides in AD progression is not yet fully understood and needs further investigation.

Besides gangliosides, we observed increased phosphatidylinositols at plaque sites in the APPPS1 brain. Altered phosphatidylinositol levels have already been reported in association with other mouse models and human AD samples [15,16]. Interestingly, the observed PI (18:0/20:4) species yielded the observed lysophosphoinositol (LPI 18:0) and arachidonic acid (AA) upon cytosolic phospholipase A2 (cPLA2) cleavage. This enzyme (cPLA2) was previously found to be more active in both mouse models and human brains [60]. Its cleavage product, arachidonic acid (AA), is metabolically important for the synthesis of mediators of inflammation [61]. This leads to the hypothesis that increased levels of PI and LPI correlate with increased inflammatory activity in the brain and are therefore also connected with microglia activation [62].

Besides the accumulation patterns of different lipid species, MSI studies on human patients with a mutation for the familial form of AD have reported the depletion of sulfatides at amyloid plaque sites [16]. An LCMS study using cell cultures and mouse and human AD patient samples confirmed this finding and showed a 30% decrease in sulfatides [24]. Sulfatides are gaining attention as they are important lipids in the central nervous system and are major components of the myelin sheath [24]. In our study, we also detected two sulfatides that appeared to be depleted at plaque sites based on the ion images. However, the statistical analysis revealed no significant depletion. It should be considered that this analysis was conducted on a total of eight comparable areas for a better comparison between the APPPS1 and WT mice. Single-plaque-level analysis could potentially yield different results. Nonetheless, comparing the abundance of lipids between PS1 and WT tissue at the single plaque level remains challenging, as there are no plaque-like regions in the WT itself.

In conclusion, this study demonstrated the level of molecular complexity in an APPPS1 mouse brain sample that could be revealed by our workflow on a single tissue section with a benchtop mass spectrometer. Besides Aβ peptides, numerous different molecules are present in the plaque microenvironment, each potentially playing a crucial role in disease progression. Therefore, it is not sufficient to consider each molecule individually; rather, they must be examined in the context of the tissue environment. This study focused on analyzing the plaque environment in a transgenic mouse model, which replicates only certain aspects of the pathological events observed in human AD patients. Although our findings align with reports from human biomarkers and lipid studies, this mouse model lacks critical pathological features such as neuronal loss and tau tangle formation. The possible impact of these missing features on lipid alterations should be considered. In addition to overexpression-based transgenic mouse models, knock-in models like APP^NL-G-F^, which introduces three mutations associated with familial AD to induce Aβ plaque formation without overexpression, may provide a more realistic representation of AD [50]. However, even these models fail to encompass key aspects such as tau pathology and neuronal loss [50]. Ultimately, studies on human tissue remain the most accurate option. Nevertheless, mouse models are still valuable because they offer unique advantages, such as enabling the long-term investigation of, e.g., drug treatment effects or genetic manipulation, which are challenging or impossible to study in humans.

In future studies of Aβ plaque metabolism, this multiomic and multimodal method may impact and be replicated with other mouse models or human AD patient samples. Only a thorough understanding of the interaction between proteins, small molecules, and different cell types will advance a more refined understanding of AD pathogenesis and reveal new drug discovery avenues.

## 4. Materials and Methods

### 4.1. Chemicals

All chemicals and solvents were of high-performance liquid chromatography–mass spectrometry (HPLC-MS) grade. Acetonitrile (ACN), acetone, water, ethanol (EtOH), sodium chloride, acetone, and 2-propanol were from VWR Chemicals (Darmstadt, Germany). Trifluoroacetic acid (TFA), α-cyano-4-hydroxycinnamic acid (α-CHCA), octyl β-D-glycopyranoside (OBG), phosphate-buffered saline (PBS), alkaline retrieval buffer, chloroform, paraformaldehyde, hydrochloric acid, and the hydrophobic barrier pen were from Merck KGaA (Darmstadt, Germany). The vacuum-stable caged MALDI matrix 4,5-dimethoxy-2-nitrobenzyl-2,5-dihydroxyacetophenone (DMNB-2,5-DHAP) [33] was provided by Sirius Fine Chemicals GmbH (SiChem, Bremen, Germany). Rabbit serum (sterile filtered) was purchased from Capricorn Scientific (Ebsdorfergrund, Germany), sodium hydroxide was from Fisher Scientific (Hampton, NH, USA), and Tris-HCl and Tris base were purchased from Carl Roth GmbH (Karlsruhe, Germany). HiPLEX Miralys “Neurology Panel” antibodies labeled with photo-cleavable mass tags (PCMTs) were obtained from AmberGen Inc. (Billerica, MA, USA).

### 4.2. Animal and Tissue Collection

APP PS1-21 (C57BL/6J-Tg(Thy1-APPSw,Thy1-PSEN1*L166P)21Jckr/J) and non-transgenic littermate (C57BL/6J) mice [7] were obtained from the Jucker lab and bred for AbbVie by Charles River Laboratories (Sulzfeld, Germany). The mice were kept in a temperature- and humidity-controlled room with a 12:12 h dark/light cycle with ad libitum access to water and food. All animal experiments were performed in full compliance with the Principles of Laboratory Animal Care [63] in an AAALAC accredited program where veterinary care and oversight were provided to ensure appropriate animal care. All animal studies were approved by the government of Rhineland Palatinate (Landesuntersuchungsamt) and conducted in accordance with the directive 2010/63/EU of the European Parliament and of the Council on the protection of animals used for scientific purpose, the ordinance on the protection of animals used for experimental or scientific purposes (German implementation of EU directive 2010/63; BGBl. I S. 3125, 3126), the Commission Recommendation 2007/526/EC on guidelines for the accommodation and care of animals used for experimental and other scientific purposes, and the German Animal Welfare Act (BGBl. I S. 1206, 1313) amended by Art. 1 G from 17 December 2018 I 2586.

Fresh-frozen brains were cut into 10 µm-thick coronal sections at −15 °C using a Leica CM 1860 UV cryostat (Leica Biosystems, Nussloch, Germany). Tissue sections were thaw-mounted onto conductive MALDI IntelliSlides™ (Part No 1868957; Bruker Daltonics GmbH, Bremen, Germany) and stored in vacuum bags at −80 °C. Prior to MS analysis, tissue sections were thawed under vacuum for approximately 1 h.

### 4.3. Sample Preparation MALDI Mass Spectrometry Imaging

For the lipid MSI experiments, tissue was spray-coated with a 2.5 mg/mL DMNB-2,5-DHAP matrix [33] in 80% ACN, 20% water, and 0.1% TFA using a TM sprayer (HTX Technologies, Chapel Hill, NC, USA) with the following spray parameters: a flow rate of 0.1 mL/min, a spray temperature of 75 °C, a nozzle height of 40 mm, ten layers sprayed in an HH pattern, and a spray velocity of 1200 mm/min.

Prior to MALDI HiPLEX-IHC staining, the MALDI matrix was removed with 2 × 3 min washes in ice-cold acetone at −80 °C and then dried under vacuum for 10 min. This was followed by the following steps in glass Coplin jars at room temperature (RT): Tissue sections were fixed in freshly prepared 1% paraformaldehyde (PFA) for 30 min, followed by a 10 min wash in PBS. Lipids and metabolites were then removed with two 3 min acetone washes followed by 3 min in Carnoy’s solution. Next, the sections were rehydrated using a series of water/EtOH washes, i.e., 2× for 2 min in 100% EtOH, 3 min in 95% EtOH, 3 min in 70% EtOH, and 3 min in 50% EtOH. After rehydration, the sections were equilibrated in 1X Tris-buffered saline (TBS) (10× stock: 50 mM Tris, 2 M NaCl, pH 7.5) for 10 min. Antigen retrieval was performed for 30 min in alkaline Tris-EDTA buffer with a pH of 9 in a water bath at 95 °C, before being equilibrated at RT for 30 min. After antigen retrieval, the sections were again washed for 10 min in TBS and blocked in tissue-blocking buffer (TBB: 2% Serum, 5% BSA in TBS with 0.05% OBG) for 1 h. PCMT-labeled antibodies were diluted in TBB and filtered (0.45 µm micro-centrifuge filter unit) for 1 min. Each tissue section on the slides was circled with a hydrophobic pen to retain small fluid volumes, and the slides were then placed in a humidity chamber. All following steps were performed in the dark. We applied 70 µL antibody probes per section, and the slides were incubated overnight at 4 °C. The slides were removed from the humidity chamber and washed 3× for 5 min in TBS with gentle shaking. Next, the sections were rebuffered for 10 s in ammonium bicarbonate (ABC) buffer, followed by rebuffering 3× for 2 min in ABC buffer with gentle shaking. After the removal of excess fluid, the sections were vacuum-dried for 1.5 h in a desiccator. Mass tags were photo-cleaved for 10 min using a near-UV Light Box (AmberGen). Next, 10 mg/mL α-CHCA in 70% ACN and 0.1% TFA were spray-coated in six passes at 60 °C, with a flow rate of 0.07 mL/min, a velocity of 1200 mm/min, a track spacing of 2 mm at a pressure of 10 psi, and a drying time of 10 s in a crisscross pattern using an HTX M5 sprayer (HTX Technologies). For the re-crystallization of the HCCA matrix, the slides were incubated in a Petri dish with a filter paper (S&S, Schleicher & Schüll, Dassel, Germany) soaked in 1 mL 5% isopropanol at 55 °C for 1 min. If not imaged immediately, the slides were stored at 4 °C in a vacuum-sealed slide mailer.

### 4.4. MALDI MS Imaging (MSI) and MS/MS Data Acquisition

The lipid MSI (*m*/*z* 400−1600; 200 shots per pixel) of coronal mouse brain sections was performed in the reflector negative ionization mode at a 20 μm pixel size using a neofleX benchtop mass spectrometer (Bruker Daltonics, Bremen, Germany) equipped with a 10 kHz smartbeam 3D laser with a 355 nm wavelength and a maximal output energy of 2 mJ. External cubic-enhanced calibration was performed using red phosphorous spotted adjacent to the tissue sections. For tandem-MS analysis, relevant lipids in the amyloid plaque region were fragmented via LID using the TOF/TOF technology of the neofleX. Precursor ions were accelerated out of the MALDI ion source and separated in the first field-free region of TOF1. Fragmentation was induced by a laser power boost of 60%. The precursor ions of interest including their already-created fragment ions were selected via an isolation unit with an isolation window of 0.65% of the respective precursor mass. The ions then reached ion source 2, where they were accelerated and focused again, followed by entering TOF2 and separation according to *m*/*z*. The remaining precursors were suppressed and the fragment ions were further focused by passing through the reflector before reaching the reflector detector. The laser ablation area of the MSMS experiments was 50 × 50 µm.

MALDI HiPLEX-IHC MSI (*m*/*z* 600–4000) was conducted in the reflector positive ionization mode at a 20 µm pixel size with a sampling rate of 1 Gs/s. Matrix suppression was set up at 640 *m*/*z*. For data processing, the smoothing algorithm was the Savitzky–Golay filter, with a width of 0.01 *m*/*z* and 5 cycles. Centroid peak detection was used with an S/N threshold of 6, a peak width of 0.05 *m*/*z*, and peak height of 80%. Baseline subtraction was performed with the TopHat algorithm.

### 4.5. Hematoxilin and Eosin (H&E) Staining

For H&E staining after lipid MSI and MALDI HiPLEX-IHC analysis, the MALDI matrix was removed by 1 min incubation in 70% ice-cold EtOH. Slides were placed in hematoxylin (Merck) for 3 min, washed with Milli-Q water for 1 min, dipped into acidic alcohol (70% EtOH with 0.1% HCl), and then washed again with Milli-Q water for 1 min. To increase the pH, a blueing step was performed (blue solution stock: 10 g Na_4_HCO_3_ + 100 g magnesium sulfate in 1 L of Milli-Q water) for 2 min, and then the slides were rinsed with Milli-Q water for 1 min. Then, the slides were counterstained in 0.5% eosin (Merck) for 2 min and washed in Milli-Q water for 1 min. The sections were finally washed and dehydrated in 80% EtOH, 96% EtOH, and 100% EtOH for 1 min each. Finally, the tissue sections were dipped into xylene (Merck) for a few seconds and then mounted with Eukitt (Merck) before microscopic images were acquired using an AperioCS2 scanner (Leica Biosystems).

### 4.6. Data Analysis

MALDI HiPLEX-IHC data were visualized in SCiLS Scope 1.0 software (Bruker Daltonics). Individual channels for all PCMTs were displayed and evaluated (Table 1). The image registration of the MALDI lipid MSI and MALDI HiPLEX-IHC was performed using in-house M^2^aia software [34,35]. Raw spectra were imported into SCiLS Lab (Bruker Daltonics).

## Figures and Tables

**Figure 1 pharmaceuticals-18-00252-f001:**
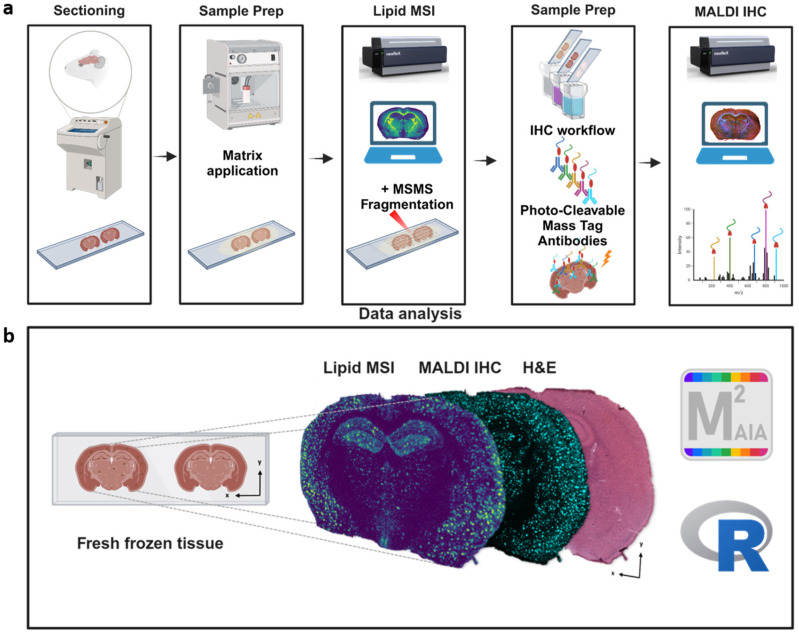
Benchtop mass spectrometry workflow for the multimodal MALDI lipid imaging, MALDI HiPLEX immunohistochemistry (IHC), and hematoxylin and eosin (H&E) histology of the Aβ plaque microenvironment in APPPS1 mice. (**a**) Fresh-frozen brains from APPPS1 and wild type (WT) mice were cryosectioned, mounted on conductive ITO glass slides, and spray-coated with the vacuum-stable MALDI matrix DMNB-2,5-DHAP [33]. Lipid MSI data were acquired on a neofleX MALDI-TOF benchtop mass spectrometer from full brain sections in the negative ionization mode. The MALDI matrix was then removed. For MALDI HiPLEX-IHC, the brain section was incubated with a mix of antibodies, each conjugated to a distinct PCMT as a reporter for a single antigen/protein marker. A second “protein” MSI run imaged the reporter mass tags on the same tissue section in the positive ionization mode. The neofleX TOF/TOF capability was used to obtain MS2 fragment spectra for a predefined list of *m*/*z* features, namely lipids, for structure elucidation on a consecutive tissue section. (**b**) Individual ion images of PCMTs were visualized in SCiLS Scope software, and the co-registration of the lipid, protein, and H&E modality layers was performed in M^2^aia software [34,35].

**Figure 2 pharmaceuticals-18-00252-f002:**
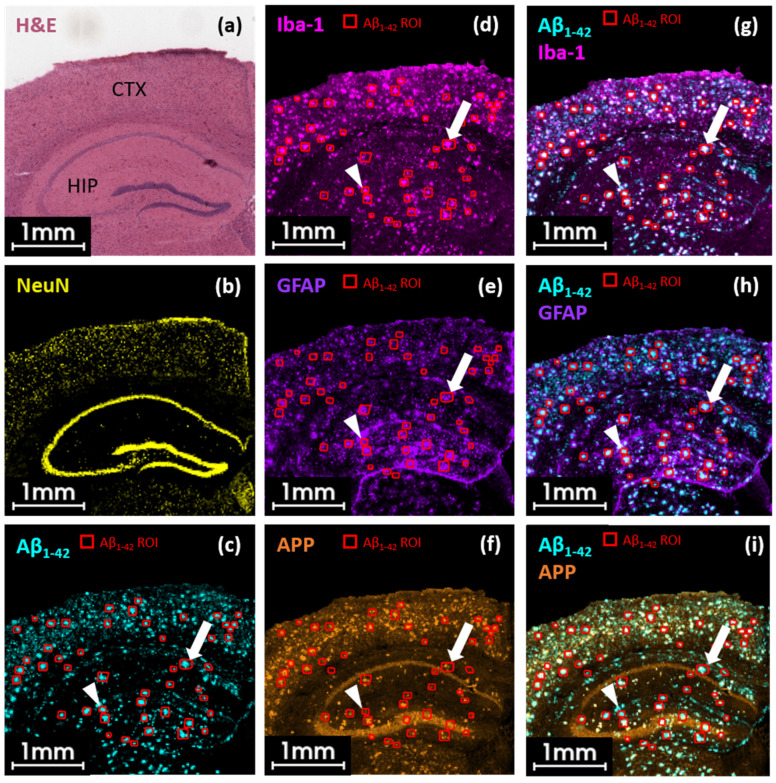
MALDI HiPLEX-IHC staining indicates the extensive co-localization of Aβ_1–42_ plaques with microglia, activated astrocytes, and the amyloid precursor protein APP in the APPPS1 mouse brain. MALDI HiPLEX-IHC images of five PCMTs indicating neuronal marker distribution in a coronal APPPS1 mouse brain section. Mass reporters of the brain structural marker NeuN (neuronal nuclei, *m*/*z* 1309.0 [M+H]^+^), amyloid-β_1–42_ peptide (Aβ_1–42_, *m*/*z* 1771.5 [M+H]^+^), the microglia marker Iba-1 (*m*/*z* 960.1 [M+H]^+^), glial fibrillary acidic protein (GFAP, *m*/*z* 1011.9 [M+H]^+^), and amyloid precursor protein (APP, *m*/*z* 1723.6 [M+H]^+^). Data were acquired in the reflector positive mode at a 20 *μ*m lateral step-size using a neofleX MALDI-TOF benchtop instrument. H&E staining (**a**) highlights distinct brain structures, e.g., the hippocampus (HIP) and cortex (CTX). H&E staining is reflected by the structural marker NeuN (yellow, (**b**)), indicating regions with a high density of neuronal cell bodies like the pyramidal neuron layer. A single ion image showing the spatial distribution of Aβ_1–42_-containing plaques (**c**), Iba-1-positive microglia (**d**), the astrogliosis marker GFAP (**e**), and amyloid precursor protein (APP, (**f**)). Fifty Aβ_1–42_ ROIs containing plaques were manually selected. Overlays of Aβ_1–42_ with Iba-1 (**g**), GFAP (**h**), and APP (**i**) show partial co-localization (white). The filled white arrow shows the co-localization of Aβ_1–42_ with all other biomarkers. The filled white arrowhead shows Aβ_1–42_ co-localization with Iba-1 and GFAP but not with APP.

**Figure 3 pharmaceuticals-18-00252-f003:**
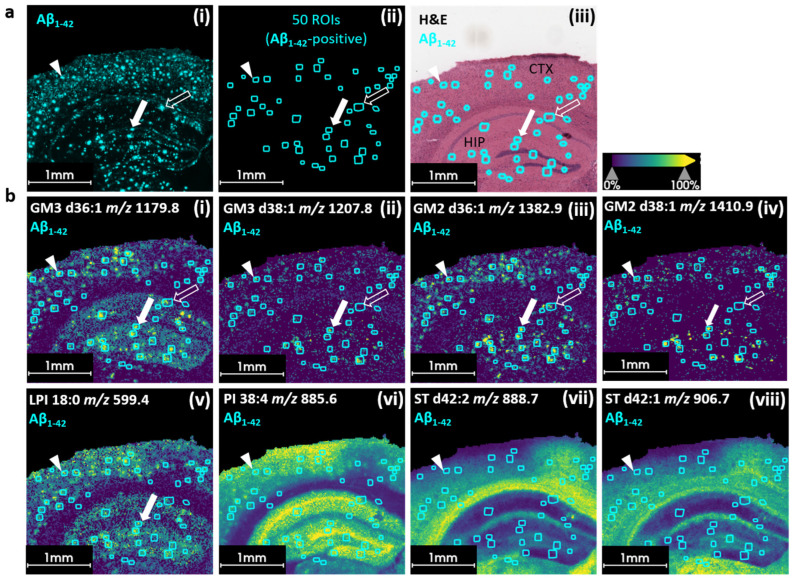
Multimodal MSI reveals lipid composition of individual Aβ_1–42_-positive plaque regions in APPPS1 mouse brain tissue. (**a**) Ion images of reporter mass for Aβ_1–42_ (*m*/*z* 1771.5 [M+H]^+^) showing general plaque distribution in cortex and hippocampus of APPPS1 mouse brain (**i**). Fifty Aβ_1–42_ ROIs containing plaques (from Figure 2) are displayed (**ii**). Overlay of fifty Aβ_1–42_-positive ROIs with H&E image of same tissue section (**iii**). (**b**) Overlays of ion images of different lipid features in negative ionization mode with fifty Aβ_1–42_-positive ROIs. (**i**–**iv**) Distributions of different GM3 and GM2 gangliosides in APPPS1 mouse brain at plaque sites. White filled arrow indicates Aβ_1–42_-positive ROIs in (**a**) as well as lysophosphatidyl inositol (LPI 18:0, *m*/*z* 599.4, [M-H]^−^) accumulation in (**b**). (**i**–**v**). White unfilled arrows indicate lipid deposition which is apparent for GM3 d36:1 (*m*/*z* 1179.8, [M-H]^−^) and GM2 d36:1 (*m*/*z* 1382.9, [M-H]^−^) but not for GM3 d38:1 (*m*/*z* 1207.8, [M-H]^−^) and GM2 d38:1 (*m*/*z* 1410.9, [M-H]^−^), thus revealing possible differences in lipid composition surrounding plaques. (**i**–**vi**) White arrow heads indicate accumulation of gangliosides and (lyso)-phosphatidylinositol as well as (**vii**,**viii**) potential depletion of two sulfatides (ST d42:2, *m*/*z* 888.7, [M-H]^−^ and ST d42:1, *m*/*z* 906.7, [M-H]^−^) in plaque environment.

**Figure 4 pharmaceuticals-18-00252-f004:**
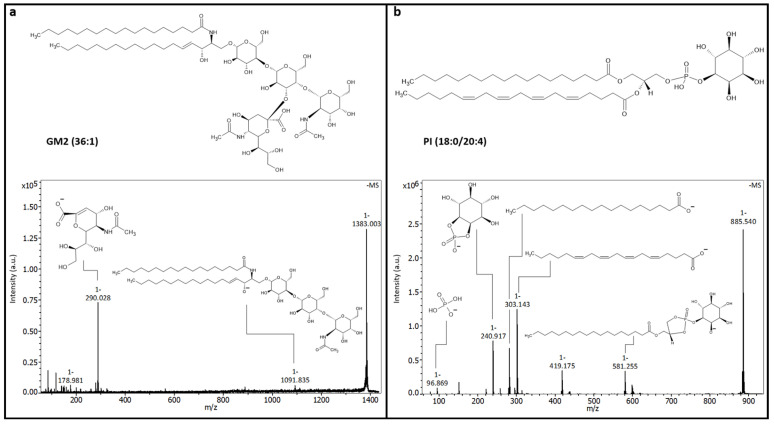
Identification of candidate lipid biomarkers by on-tissue TOF/TOF fragmentation analysis using benchtop mass spectrometer. (**a**) Structure of ganglioside GM2 (36:1) (C_67_H_121_N_3_O_26_; *m*/*z* 1383.8 [M]) and MS^2^ fragmentation spectrum of *m*/*z* 1383.0 [M-H]^−^ in negative ionization mode obtained on neofleX TOF/TOF instrument. (**b**) Structure of phosphatidylinositol 38:4 (18:0/20:4) (C_47_H_83_O_13_P; *m*/*z* 885.5 [M]) and MS^2^ fragmentation spectrum of *m*/*z* 885.5 [M-H]^−^ in negative ionization mode.

**Table 1 pharmaceuticals-18-00252-t001:** Theoretical masses of photo-cleavable and mass-tagged (PCMT) antibodies and their corresponding antigens and concentrations.

Antigen	PCMT Theoretical Mass	Concentration [µg/mL]
Iba-1	959.57	3.75
Amyloid-β_1–42_ (Aβ_1–42_)	1770.87	3.75
GFAP	1011.54	3.75
NeuN	1308.70	3.75
APP	1722.93	5.00

## Data Availability

Data will be available from the corresponding author after publication upon reasonable request.

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
