# Peer review of "Exploring the Aβ Plaque Microenvironment in Alzheimer’s Disease Model Mice by Multimodal Lipid-Protein-Histology Imaging on a Benchtop Mass Spectrometer"

_pharmaceuticals, 2025, doi:10.3390/ph18020252_

Round 1
Reviewer 1 Report
Comments and Suggestions for Authors
Comments to authors
This manuscript presents an innovative multimodal approach to study Alzheimer's disease (AD) by combining MALDI mass spectrometry imaging (MSI), MALDI HiPLEX immunohistochemistry (IHC), and histological techniques. The study focused on the microenvironment of Aβ1-42 plaques in an AD mouse model (APP/PS1), examining lipid composition and immune markers such as Iba-1, GFAP, and APP. Major findings include the identification of specific lipid species (e.g., GM2, PI 38:4) in the vicinity of plaques, elucidating the role of lipid metabolism in AD progression. The use of a benchtop mass spectrometer is a novel and practical approach that provides a convenient, cost-effective alternative to high-end instrumentation. The strength of the study lies in its integrated analysis, which provides a comprehensive view of molecular interactions in AD. The method is exhaustive and ensures the reliability and reproducibility of the data, making it a significant contribution to AD research. Therefore, I recommend minor revisions to the manuscript. My comments are as follows:
Minor comments
1. Lack of quantitative data: The manuscript provides qualitative analysis of lipid accumulation and protein colocalization, but lacks quantitative data to support these findings. For example, the degree of colocalization between Aβ1-42 plaques and immune markers (Iba-1, GFAP) is described, but no statistical analysis is provided to confirm the results. Including quantitative data, such as the percentage of colocalized areas or the relative abundance of lipid species, would strengthen the manuscript and increase the reliability of the findings.
2. Limited discussion on model specificity: While the APP/PS1 mouse model is widely used in AD research, it primarily represents early-onset familial AD and may not fully capture the complexity of late-onset sporadic AD. The manuscript briefly mentions this limitation, but a more detailed discussion of how the results may differ in other AD models or human patients with sporadic AD would be helpful. In addition, the absence of GM1 accumulation in plaques should be further explored, especially in comparison with other AD models.
3. No direct comparison with human samples: While the results from the mouse model are promising, there is no direct comparison with human AD samples. Integration of data from human AD tissues would validate the findings and improve the clinical relevance of the study. This could be a limitation if the goal is to translate the findings to human AD pathology.
4. References: The references are relatively old, with most of them older than the past five years.
5. Incomplete analysis of lipid changes: Although the study provides valuable insights into lipid changes surrounding Aβ plaques, a more thorough analysis of lipid metabolism would benefit. For example, the manuscript mentions depletion of sulfate esters, but the effects of this depletion are not discussed in detail. Potential links between lipid degradation and Aβ plaque formation, as well as the role of lysosomal activity in lipid metabolism, could be further explored.
6. Lack of statistical validation of lipid analysis: The manuscript does not provide statistical validation of the lipid profiles and the differences observed between APP/PS1 and wild-type (WT) mice. Including statistical analyses (e.g., p-values, confidence intervals) to support the observed lipid changes would improve the robustness of the findings.
Author Response
Response letter for Reviewer 1
We thank the Reviewer very much for the thoughtful and encouraging comments on our manuscript. We are delighted that our multimodal approach for studying Alzheimer’s disease (AD) model mice was found to be innovative and impactful, particular the integration of MALDI mass spectrometry imaging, MALDI HiPLEX immunohistochemistry, and histological staining to investigate the microenvironment of Aβ1-42 plaques in the APPPS1 mouse model. We are also glad that the identification of specific lipid species, such as GM2 and PI 38:4, were convincing key findings that highlight the complex role of lipid metabolism in AD progression. The recognition of the practical benefits of using a benchtop mass spectrometer as a cost-effective alternative to high-end instrumentation is greatly appreciated and reinforces our goal of making advanced methodologies like MALDI imaging more accessible. The acknowledgment of the comprehensive nature of our integrated analysis and its contribution to the reliability and reproducibility of the data is truly valued.
Comments to authors
This manuscript presents an innovative multimodal approach to study Alzheimer's disease (AD) by combining MALDI mass spectrometry imaging (MSI), MALDI HiPLEX immunohistochemistry (IHC), and histological techniques. The study focused on the microenvironment of Aβ1-42 plaques in an AD mouse model (APP/PS1), examining lipid composition and immune markers such as Iba-1, GFAP, and APP. Major findings include the identification of specific lipid species (e.g., GM2, PI 38:4) in the vicinity of plaques, elucidating the role of lipid metabolism in AD progression. The use of a benchtop mass spectrometer is a novel and practical approach that provides a convenient, cost-effective alternative to high-end instrumentation. The strength of the study lies in its integrated analysis, which provides a comprehensive view of molecular interactions in AD. The method is exhaustive and ensures the reliability and reproducibility of the data, making it a significant contribution to AD research. Therefore, I recommend minor revisions to the manuscript. My comments are as follows:
Minor comments
- Lack of quantitative data: The manuscript provides qualitative analysis of lipid accumulation and protein colocalization, but lacks quantitative data to support these findings. For example, the degree of colocalization between Aβ1-42 plaques and immune markers (Iba-1, GFAP) is described, but no statistical analysis is provided to confirm the results. Including quantitative data, such as the percentage of colocalized areas or the relative abundance of lipid species, would strengthen the manuscript and increase the reliability of the findings.
We agree that including statistical analysis to compare the relative abundances of lipid species between APPPS1 and WT mice, as well as providing more detailed data analysis for the degree of co-localization between the different markers would add significant value to the conclusions.
Changes: Therefore, we added an evaluation of co-localization between Aβ1-42 plaques and immune markers (Iba-1, GFAP, APP) (see Supplementary figure 2). In addition, we added a statistical evaluation of the lipid candidate accumulation (see Supplementary figure 4).
- Limited discussion on model specificity: While the APP/PS1 mouse model is widely used in AD research, it primarily represents early-onset familial AD and may not fully capture the complexity of late-onset sporadic AD. The manuscript briefly mentions this limitation, but a more detailed discussion of how the results may differ in other AD models or human patients with sporadic AD would be helpful. In addition, the absence of GM1 accumulation in plaques should be further explored, especially in comparison with other AD models.
We greatly appreciate the comments regarding the limitations of the APPPS1 mouse model. We fully agree that there are limitations, and we do not believe that the results obtained from this model can be directly translated to other mouse models or human AD patient samples.
Changes: To address this, we have expanded the discussion to include a more detailed view on how the findings might differ in mouse models using different approaches to APP overexpression (like in the APP NL-G-F Knock-in mouse model) or human AD patient samples.
We also had a deeper look into the spatial distribution of GM1 in our results and compared those to already published data.
Changes: We have expanded the discussion about this GM1 observation in the respective discussion section.
- No direct comparison with human samples: While the results from the mouse model are promising, there is no direct comparison with human AD samples. Integration of data from human AD tissues would validate the findings and improve the clinical relevance of the study. This could be a limitation if the goal is to translate the findings to human AD pathology.
We thank the reviewer for this comment and we fully agree that the integration of human data would be an important next step. However, obtaining human data is out of scope for this study, as we aimed to show a new workflow on a single slide and that low cost alternatives (benchtop mass spectrometers) can be used for advanced and highly integrative data analysis in AD research. Still we added a more in-depth discussion comparing our data to already existing human data from AD studies.
Changes: We added additional information and references in the discussion section about the findings in human AD studies.
- References: The references are relatively old, with most of them older than the past five years.
We thank the reviewer for pointing this out.
Changes: We have gone through the text again and exchanged many older references with more recent publications.
- Incomplete analysis of lipid changes: Although the study provides valuable insights into lipid changes surrounding Aβ plaques, a more thorough analysis of lipid metabolism would benefit. For example, the manuscript mentions depletion of sulfate esters, but the effects of this depletion are not discussed in detail. Potential links between lipid degradation and Aβ plaque formation, as well as the role of lysosomal activity in lipid metabolism, could be further explored.
We appreciate the comment on a more detailed lipid metabolism analysis.
Changes: We included a more detailed analysis of possible alterations in lipid metabolism in the discussion section.
- Lack of statistical validation of lipid analysis: The manuscript does not provide statistical validation of the lipid profiles and the differences observed between APP/PS1 and wild-type (WT) mice. Including statistical analyses (e.g., p-values, confidence intervals) to support the observed lipid changes would improve the robustness of the findings.
Response letter for Reviewer 1
We thank the Reviewer very much for the thoughtful and encouraging comments on our manuscript. We are delighted that our multimodal approach for studying Alzheimer’s disease (AD) model mice was found to be innovative and impactful, particular the integration of MALDI mass spectrometry imaging, MALDI HiPLEX immunohistochemistry, and histological staining to investigate the microenvironment of Aβ1-42 plaques in the APPPS1 mouse model. We are also glad that the identification of specific lipid species, such as GM2 and PI 38:4, were convincing key findings that highlight the complex role of lipid metabolism in AD progression. The recognition of the practical benefits of using a benchtop mass spectrometer as a cost-effective alternative to high-end instrumentation is greatly appreciated and reinforces our goal of making advanced methodologies like MALDI imaging more accessible. The acknowledgment of the comprehensive nature of our integrated analysis and its contribution to the reliability and reproducibility of the data is truly valued.
Comments to authors
This manuscript presents an innovative multimodal approach to study Alzheimer's disease (AD) by combining MALDI mass spectrometry imaging (MSI), MALDI HiPLEX immunohistochemistry (IHC), and histological techniques. The study focused on the microenvironment of Aβ1-42 plaques in an AD mouse model (APP/PS1), examining lipid composition and immune markers such as Iba-1, GFAP, and APP. Major findings include the identification of specific lipid species (e.g., GM2, PI 38:4) in the vicinity of plaques, elucidating the role of lipid metabolism in AD progression. The use of a benchtop mass spectrometer is a novel and practical approach that provides a convenient, cost-effective alternative to high-end instrumentation. The strength of the study lies in its integrated analysis, which provides a comprehensive view of molecular interactions in AD. The method is exhaustive and ensures the reliability and reproducibility of the data, making it a significant contribution to AD research. Therefore, I recommend minor revisions to the manuscript. My comments are as follows:
Minor comments
- Lack of quantitative data: The manuscript provides qualitative analysis of lipid accumulation and protein colocalization, but lacks quantitative data to support these findings. For example, the degree of colocalization between Aβ1-42 plaques and immune markers (Iba-1, GFAP) is described, but no statistical analysis is provided to confirm the results. Including quantitative data, such as the percentage of colocalized areas or the relative abundance of lipid species, would strengthen the manuscript and increase the reliability of the findings.
We agree that including statistical analysis to compare the relative abundances of lipid species between APPPS1 and WT mice, as well as providing more detailed data analysis for the degree of co-localization between the different markers would add significant value to the conclusions.
Changes: Therefore, we added an evaluation of co-localization between Aβ1-42 plaques and immune markers (Iba-1, GFAP, APP) (see Supplementary figure 2). In addition, we added a statistical evaluation of the lipid candidate accumulation (see Supplementary figure 4).
- Limited discussion on model specificity: While the APP/PS1 mouse model is widely used in AD research, it primarily represents early-onset familial AD and may not fully capture the complexity of late-onset sporadic AD. The manuscript briefly mentions this limitation, but a more detailed discussion of how the results may differ in other AD models or human patients with sporadic AD would be helpful. In addition, the absence of GM1 accumulation in plaques should be further explored, especially in comparison with other AD models.
We greatly appreciate the comments regarding the limitations of the APPPS1 mouse model. We fully agree that there are limitations, and we do not believe that the results obtained from this model can be directly translated to other mouse models or human AD patient samples.
Changes: To address this, we have expanded the discussion to include a more detailed view on how the findings might differ in mouse models using different approaches to APP overexpression (like in the APP NL-G-F Knock-in mouse model) or human AD patient samples.
We also had a deeper look into the spatial distribution of GM1 in our results and compared those to already published data.
Changes: We have expanded the discussion about this GM1 observation in the respective discussion section.
- No direct comparison with human samples: While the results from the mouse model are promising, there is no direct comparison with human AD samples. Integration of data from human AD tissues would validate the findings and improve the clinical relevance of the study. This could be a limitation if the goal is to translate the findings to human AD pathology.
We thank the reviewer for this comment and we fully agree that the integration of human data would be an important next step. However, obtaining human data is out of scope for this study, as we aimed to show a new workflow on a single slide and that low cost alternatives (benchtop mass spectrometers) can be used for advanced and highly integrative data analysis in AD research. Still we added a more in-depth discussion comparing our data to already existing human data from AD studies.
Changes: We added additional information and references in the discussion section about the findings in human AD studies.
- References: The references are relatively old, with most of them older than the past five years.
We thank the reviewer for pointing this out.
Changes: We have gone through the text again and exchanged many older references with more recent publications.
- Incomplete analysis of lipid changes: Although the study provides valuable insights into lipid changes surrounding Aβ plaques, a more thorough analysis of lipid metabolism would benefit. For example, the manuscript mentions depletion of sulfate esters, but the effects of this depletion are not discussed in detail. Potential links between lipid degradation and Aβ plaque formation, as well as the role of lysosomal activity in lipid metabolism, could be further explored.
We appreciate the comment on a more detailed lipid metabolism analysis.
Changes: We included a more detailed analysis of possible alterations in lipid metabolism in the discussion section.
- Lack of statistical validation of lipid analysis: The manuscript does not provide statistical validation of the lipid profiles and the differences observed between APP/PS1 and wild-type (WT) mice. Including statistical analyses (e.g., p-values, confidence intervals) to support the observed lipid changes would improve the robustness of the findings.
We thank the reviewer for highlighting the lack of statistical analysis validation of the lipid candidates. We included a statistical analysis of lipids in the Supplement performing unpaired t-tests to validate the visual impressions of the ion-images.
Changes: We added a figure in the supplement (Supplementary Figure 4) and included the statistical outcome in the main text (Results; “2.3. Multimodal MSI reveals lipid composition of individual Aβ1-42-positive microglia regions in AD mouse APPPS1 mouse brain tissue”).
Response letter for Reviewer 1
We thank the Reviewer very much for the thoughtful and encouraging comments on our manuscript. We are delighted that our multimodal approach for studying Alzheimer’s disease (AD) model mice was found to be innovative and impactful, particular the integration of MALDI mass spectrometry imaging, MALDI HiPLEX immunohistochemistry, and histological staining to investigate the microenvironment of Aβ1-42 plaques in the APPPS1 mouse model. We are also glad that the identification of specific lipid species, such as GM2 and PI 38:4, were convincing key findings that highlight the complex role of lipid metabolism in AD progression. The recognition of the practical benefits of using a benchtop mass spectrometer as a cost-effective alternative to high-end instrumentation is greatly appreciated and reinforces our goal of making advanced methodologies like MALDI imaging more accessible. The acknowledgment of the comprehensive nature of our integrated analysis and its contribution to the reliability and reproducibility of the data is truly valued.
Comments to authors
This manuscript presents an innovative multimodal approach to study Alzheimer's disease (AD) by combining MALDI mass spectrometry imaging (MSI), MALDI HiPLEX immunohistochemistry (IHC), and histological techniques. The study focused on the microenvironment of Aβ1-42 plaques in an AD mouse model (APP/PS1), examining lipid composition and immune markers such as Iba-1, GFAP, and APP. Major findings include the identification of specific lipid species (e.g., GM2, PI 38:4) in the vicinity of plaques, elucidating the role of lipid metabolism in AD progression. The use of a benchtop mass spectrometer is a novel and practical approach that provides a convenient, cost-effective alternative to high-end instrumentation. The strength of the study lies in its integrated analysis, which provides a comprehensive view of molecular interactions in AD. The method is exhaustive and ensures the reliability and reproducibility of the data, making it a significant contribution to AD research. Therefore, I recommend minor revisions to the manuscript. My comments are as follows:
Minor comments
- Lack of quantitative data: The manuscript provides qualitative analysis of lipid accumulation and protein colocalization, but lacks quantitative data to support these findings. For example, the degree of colocalization between Aβ1-42 plaques and immune markers (Iba-1, GFAP) is described, but no statistical analysis is provided to confirm the results. Including quantitative data, such as the percentage of colocalized areas or the relative abundance of lipid species, would strengthen the manuscript and increase the reliability of the findings.
We agree that including statistical analysis to compare the relative abundances of lipid species between APPPS1 and WT mice, as well as providing more detailed data analysis for the degree of co-localization between the different markers would add significant value to the conclusions.
Changes: Therefore, we added an evaluation of co-localization between Aβ1-42 plaques and immune markers (Iba-1, GFAP, APP) (see Supplementary figure 2). In addition, we added a statistical evaluation of the lipid candidate accumulation (see Supplementary figure 4).
- Limited discussion on model specificity: While the APP/PS1 mouse model is widely used in AD research, it primarily represents early-onset familial AD and may not fully capture the complexity of late-onset sporadic AD. The manuscript briefly mentions this limitation, but a more detailed discussion of how the results may differ in other AD models or human patients with sporadic AD would be helpful. In addition, the absence of GM1 accumulation in plaques should be further explored, especially in comparison with other AD models.
We greatly appreciate the comments regarding the limitations of the APPPS1 mouse model. We fully agree that there are limitations, and we do not believe that the results obtained from this model can be directly translated to other mouse models or human AD patient samples.
Changes: To address this, we have expanded the discussion to include a more detailed view on how the findings might differ in mouse models using different approaches to APP overexpression (like in the APP NL-G-F Knock-in mouse model) or human AD patient samples.
We also had a deeper look into the spatial distribution of GM1 in our results and compared those to already published data.
Changes: We have expanded the discussion about this GM1 observation in the respective discussion section.
- No direct comparison with human samples: While the results from the mouse model are promising, there is no direct comparison with human AD samples. Integration of data from human AD tissues would validate the findings and improve the clinical relevance of the study. This could be a limitation if the goal is to translate the findings to human AD pathology.
We thank the reviewer for this comment and we fully agree that the integration of human data would be an important next step. However, obtaining human data is out of scope for this study, as we aimed to show a new workflow on a single slide and that low cost alternatives (benchtop mass spectrometers) can be used for advanced and highly integrative data analysis in AD research. Still we added a more in-depth discussion comparing our data to already existing human data from AD studies.
Changes: We added additional information and references in the discussion section about the findings in human AD studies.
- References: The references are relatively old, with most of them older than the past five years.
We thank the reviewer for pointing this out.
Changes: We have gone through the text again and exchanged many older references with more recent publications.
- Incomplete analysis of lipid changes: Although the study provides valuable insights into lipid changes surrounding Aβ plaques, a more thorough analysis of lipid metabolism would benefit. For example, the manuscript mentions depletion of sulfate esters, but the effects of this depletion are not discussed in detail. Potential links between lipid degradation and Aβ plaque formation, as well as the role of lysosomal activity in lipid metabolism, could be further explored.
We appreciate the comment on a more detailed lipid metabolism analysis.
Changes: We included a more detailed analysis of possible alterations in lipid metabolism in the discussion section.
- Lack of statistical validation of lipid analysis: The manuscript does not provide statistical validation of the lipid profiles and the differences observed between APP/PS1 and wild-type (WT) mice. Including statistical analyses (e.g., p-values, confidence intervals) to support the observed lipid changes would improve the robustness of the findings.
We thank the reviewer for highlighting the lack of statistical analysis validation of the lipid candidates. We included a statistical analysis of lipids in the Supplement performing unpaired t-tests to validate the visual impressions of the ion-images.
Changes: We added a figure in the supplement (Supplementary Figure 4) and included the statistical outcome in the main text (Results; “2.3. Multimodal MSI reveals lipid composition of individual Aβ1-42-positive microglia regions in AD mouse APPPS1 mouse brain tissue”).
We thank the reviewer for highlighting the lack of statistical analysis validation of the lipid candidates. We included a statistical analysis of lipids in the Supplement performing unpaired t-tests to validate the visual impressions of the ion-images.
Changes: We added a figure in the supplement (Supplementary Figure 4) and included the statistical outcome in the main text (Results; “2.3. Multimodal MSI reveals lipid composition of individual Aβ1-42-positive microglia regions in AD mouse APPPS1 mouse brain tissue”).
Reviewer 2 Report
Comments and Suggestions for Authors
This manuscript examines the microenvironment round A-beta plaques in a mouse model of Alzheimer’s disease. It has several shortcomings:
1. The authors use the APP/PS1-21 mouse model of Alzheimer's disease (AD). This model carries human transgenes for amyloid precursor protein (APP) with the Swedish mutation (K670N/M671L) and presenilin-1 (PS1) with the L166P mutation. Therefore, not only is it incorrect to use the terms ‘AD mice’ and ‘AD tissue sample’, the actual relevance of this study to AD itself can be questioned.
2. The Introduction needs to indicate that the third key pathological feature is the loss of neurons.
3. The authors should refer to the three monoclonal antibodies against A-beta that have been approved by the FDA for the treatment of AD.
4. The legend to figure 2 does not include a description over the 9 image is presented. This needs to be corrected.
5. Selection to figure 3 also needs to be corrected along similar lines.
6. No statistical analysis seems to have been performed. Therefore. any indication of change in the manuscript is inappropriate.
Comments on the Quality of English LanguageOK
Author Response
Response letter Revision Reviewer 2
We thank the Reviewer for taking the time to review our manuscript on using a multimodal approach to study pathological features known from Alzheimer’s disease in the transgenic mouse model APPPS1. We appreciate the thoughtful and critical comments.
Comments and Suggestions for Authors
This manuscript examines the microenvironment round A-beta plaques in a mouse model of Alzheimer’s disease. It has several shortcomings:
- 1. The authors use the APP/PS1-21 mouse model of Alzheimer's disease (AD). This model carries human transgenes for amyloid precursor protein (APP) with the Swedish mutation (K670N/M671L) and presenilin-1 (PS1) with the L166P mutation. Therefore, not only is it incorrect to use the terms ‘AD mice’ and ‘AD tissue sample’, the actual relevance of this study to AD itself can be questioned.
We appreciate the critical comment on our use of the terms “AD mice” or “AD tissue” in this manuscript. It is correct that this transgenic mouse model exhibits Alzheimer’s-like features but does not fully represent the complete AD pathology.
Changes: We added precise information on the transgenes to the method section and replaced incorrect terms within the manuscript.
- 2. The Introduction needs to indicate that the third key pathological feature is the loss of neurons.
Changes: We revised the introduction and included the information about the loss of neurons as a key pathological feature in the progression of Alzheimer’s disease (Introduction section).
- 3. The authors should refer to the three monoclonal antibodies against A-beta that have been approved by the FDA for the treatment of AD.
We welcome the comment on the FDA-approved AD drugs on the market.
Changes: We included this information in the introduction of the manuscript (Introduction section).
- The legend to figure 2 does not include a description over the 9 image is presented. This needs to be corrected.
We appreciate the reviewer bringing to our attention that we did not reference the correct figure and supplementary figure for the reported results of the MALDI HiPLEX-IHC staining.
Changes: We updated the main text in the results part (Chapter: 2.2. MALDI HiPLEX-IHC staining reveals AD biomarker co-localization in the microenvironment of Aβ1-42-positive plaques in APPPS1 mouse brain) and referred to the correct figures and supplementary figures.
- Selection to figure 3 also needs to be corrected along similar lines.
We also very much appreciate the reviewer bringing to our attention that we did not reference the correct figure and supplementary figure for the reported results of the MALDI MSI lipid data.
Changes: We updated the main text in the results part (Chapter: Multimodal MSI reveals lipid composition of individual Aβ1-42-positive microglia regions in APPPS1 mouse brain tissue) and referred to the correct figures and supplementary figures.
- No statistical analysis seems to have been performed. Therefore, any indication of change in the manuscript is inappropriate.
We agree that including statistical analysis to compare the relative abundances of lipid species between APPPS1 and WT mice will add significant value to the conclusions.
Changes: We added a statistical evaluation of the lipid candidate accumulation (see Supplementary Figure 4).
Reviewer 3 Report
Comments and Suggestions for Authors
MALDI MSI is a valuable tool for analyzing amyloid plaque deposits, offering unique insights into their composition, distribution, and aggregation states. Its application in both basic research and clinical studies holds promise for advancing our understanding of Alzheimer’s disease and related disorders. As technological innovations address current limitations, MALDI MSI will likely play an increasingly important role in unraveling the molecular mechanisms underlying amyloid pathology and aiding in the development of effective diagnostics and therapeutics. The authors successfully demonstrated the application of this technology for the detection of lipids, protein markers, and hematoxylin and eosin histology on the same tissue section. For these points I consider the manuscript worthy of publication. However some additional discussion is required:
1. A detailed description of the MALDI-MSI experimental conditions is essential to help readers fully understand the workflow and appreciate the enhanced capabilities of the new instrument used in this study. Please add information concerning type of the laser, its energy levels used in the experiments, technology of tandem experiments etc.
2. More detailed description of the procedure for the identification of candidate lipid biomarkers is required. Please mark ions used for the determination of FAs in ceramides fragments of gangliosides. Add some details concerning determination of double bond positions in FAs.
Author Response
Response letter Revision Reviewer 3
We would like to express our sincere gratitude to the Reviewer for his insightful and encouraging feedback on our work. We are delighted that the potential of MALDI MSI as a powerful tool for studying amyloid plaque deposits and its broader implications in advancing research on Alzheimer's disease has been acknowledged. We truly appreciate the recognition of the technology's unique ability to provide insights into plaque composition, distribution, and aggregation states. The comment regarding the technology's growing importance, as evidenced by technological advancements, aligns with our vision for future applications of MALDI MSI. We are gratified to learn that our demonstration, which aimed to enable the detection of lipids, protein markers, and histology features on the same tissue section, was perceived to be successful.
Comments and Suggestions for Authors
MALDI MSI is a valuable tool for analyzing amyloid plaque deposits, offering unique insights into their composition, distribution, and aggregation states. Its application in both basic research and clinical studies holds promise for advancing our understanding of Alzheimer’s disease and related disorders. As technological innovations address current limitations, MALDI MSI will likely play an increasingly important role in unraveling the molecular mechanisms underlying amyloid pathology and aiding in the development of effective diagnostics and therapeutics. The authors successfully demonstrated the application of this technology for the detection of lipids, protein markers, and hematoxylin and eosin histology on the same tissue section. For these points I consider the manuscript worthy of publication. However some additional discussion is required:
- Detailed description of the MALDI-MSI experimental conditions is essential to help readers fully understand the workflow and appreciate the enhanced capabilities of the new instrument used in this study. Please add information concerning type of the laser, its energy levels used in the experiments, technology of tandem experiments etc.
We thank the reviewer for commenting on adding additional information needed for our MALDI MSI method section.
Changes: We added the requested details in the appropriated sections (Materials and Methods „MALDI MS Imaging (MSI) and MS/MS Data Acquisition”).
- More detailed description of the procedure for the identification of candidate lipid biomarkers is required. Please mark ions used for the determination of FAs in ceramides fragments of gangliosides. Add some details concerning determination of double bond positions in FAs.
We appreciate the comment on detailed description how to identify FA chains as well as the double bond positions within these fatty acids. We were able to verify the head group as well as the sum composition of the FA chains for all fragmented lipids and therefore are not reporting the lipid species notation (Liebisch et al. 2023, https://doi.org/10.1194/jlr.M033506) instead of the specific double bond position.
Changes: We made text changes in the Supplement (Supplement figure 5) and in the main text (Results, Chapter: 2.4. Structure elucidation of candidate lipid biomarkers by TOF/TOF on-tissue fragmentation using a bench top mass spectrometer) and the figure legend of Figure 4.
Round 2
Reviewer 2 Report
Comments and Suggestions for Authors
The authors have addressed my concerns